# Agreement on the Prevalence of Body Mass Index (BMI) in Mexican Children and Adolescents Using Different International References

**DOI:** 10.3390/nu17030587

**Published:** 2025-02-06

**Authors:** Marisol Santiago-Arango, Eduardo Pérez-Campos, Ivan Porras-Chaparro, Juan de Dios Ruiz-Rosado, Héctor Martínez Ruiz, Héctor A. Cabrera-Fuentes, Eduardo Pérez-Campos Mayoral, Margarito Martínez-Cruz, María Teresa Hernández-Huerta, Laura Pérez-Campos Mayoral

**Affiliations:** 1UNAM-UABJO Faculty of Medicine Research Centre, Faculty of Medicine and Surgery, Benito Juárez Autonomous University of Oaxaca, Oaxaca 68120, Mexico; stgmarisol@gmail.com (M.S.-A.); drheccctor@hotmail.com (H.M.R.); eperezcampos.fmc@uabjo.mx (E.P.-C.M.); 2National Technological of Mexico, ITOaxaca, Oaxaca 68033, Mexico; pcampos@itoaxaca.edu.mx (E.P.-C.); mcruz@itoaxaca.edu.mx (M.M.-C.); 3Faculty of Economics, Benito Juárez Autonomous University of Oaxaca, Oaxaca 68120, Mexico; iporras23@cecad-uabjo.mx; 4Kidney and Urinary Tract Research Center, Abigail Wexner Research Institute, Nationwide Children’s Hospital, Columbus, OH 43215, USA; juandedios.ruizrosado@nationwidechildrens.org; 5Division of Nephrology and Hypertension, Nationwide Children’s Hospital, Columbus, OH 43205, USA; 6R&D Group, Office of the Vice President for Scientific Research and Innovation, Imam Abdulrahman Bin Faisal University Dammam, Dammam 31441, Saudi Arabia; halexcafu@gmail.com; 7Interdisciplinary Centre for Biological and Human Research, Benito Juárez Autonomous University of Oaxaca, Oaxaca 68000, Mexico; 8Consejo Nacional de Ciencia y Tecnología (CONAHCYT), Faculty of Medicine and Surgery, Benito Juárez Autonomous University of Oaxaca, Oaxaca 68120, Mexico

**Keywords:** National Health and Nutrition Survey, child and adolescent nutrition, obesity, overweight, International Obesity Task Force, Centers for Disease Control and Prevention

## Abstract

(1) Background: Obesity and overweight are defined as an abnormal or excessive accumulation of fat that can be harmful to health. These are conditions that can lead to a lifetime of diseases, including cancer, diabetes, and heart disease. The diagnosis of overweight and obesity in children and adolescents depends on the international reference used. (2) Objectives: In this study aimed to determine the level of concordance of the prevalence of underweight, normal weight, overweight, and obesity were estimated with three international references in Mexican children and adolescents between 2 and 18 years of age. (3) Methods: We used specific the body mass index (BMI) cut-off points of the ENSANUT Continua 2022 database. The weight category was measured using the World Health Organization (WHO) criteria (<−2 standard deviation (SD), underweight; −2 and +0.99 SD, normal weight; +1 and +1.99 SD, overweight; and ≥+2 SD, obesity); Centers for Disease Control and Prevention (CDC) criteria (<percentile 5, underweight; percentile 5 and <percentile 85, normal weight; ≥percentile 85, overweight; and ≥percentile 95, obesity); and the Obesity Task Force (IOTF) criteria (specific limit values). To determine agreement among these three criteria, Cohen’s Kappa index was used. (4) Results: There were differences in the estimation of weight categories according to the international reference used. Substantial (WHO-IOTF: 0.639; CDC-IOTF: 0.785) and almost perfect (WHO-CDC: 0.806) levels of agreement were found between the references used; however, agreement varied according to age. (5) Conclusions: The weight category depends on the reference used, so each one should be used with caution since the results inform our actions of prevention, surveillance, and the control of nutrition in childhood and adolescence for the timely detection of chronic health problems and effects of social deficiencies.

## 1. Introduction

Overweight and obesity are a public health problem present in the adult population as well as in children and adolescents. Furthermore, this problem manifests in developed and developing countries [1]. According to the World Health Organization (WHO), since the 1990s, obesity in adults has doubled, and quadrupled in adolescents [2]. In 2022, 48 out of every 100 adults aged 18 and over were overweight, and 16 out of every 100 were obese. In addition, more than 390 million children and adolescents between 5 and 19 years of age were reported to be overweight, 160 million of them were obese, and 37 million children under 5 years of age were overweight [2].

In Mexico, according to data from the National Continuous Health and Nutrition Survey (ENSANUT Continua, for its Spanish acronym), during 2020–2022, the prevalence of overweight in schoolchildren was 19.2%, and that of obesity was 18.1%. In adolescents, the prevalence of overweight was 23.9%, and that of obesity was 17.2% [3,4]. The diagnosis of overweight and obesity is realized through the body mass index (BMI), that is, the relationship between weight and height (weight/height^2^). This indicator is the most common for detecting underweight, normal weight, overweight, and obesity [5]. In adults, the interpretation of the BMI is based on standard weight categories. The classification is the same for men and women, regardless of body type and age. In the case of children and adolescents, the interpretation is different. Although the calculation is the same as for adults, the BMI is specific for age and sex because the amount of body fat changes with age, in addition to being different between sexes [6].

The BMI is a simple and low-cost way of determining weight categories that can cause health problems in adults and children. The importance of establishing BMIs in children and adolescents is because if they are underweight at these stages, they may have deficits in their physical and cognitive condition and a weak immune system, which implies a greater probability of suffering infections [7]. If, on the other hand, children and adolescents are overweight or obese, they are more likely to have this condition throughout their lives, which implies the development of non-communicable diseases such as type 2 diabetes [8], insulin resistance [9], dyslipidemia, and orthopedic problems, among others [8,10]. In addition, it increases the probability of presenting discrimination and self-esteem problems [11].

Although the BMI estimate for children and adolescents is the same as for adults, it is known as the BMI for age. Internationally, different cut-off points have been established to assess the nutritional status of children and adolescents based on the BMI, which is favored by the absence of clear points to determine weight class [1]. In this sense, organizations such as the WHO [12,13], Centers for Disease Control and Prevention (CDC) [14], and the Word Obesity Federation (WOF) [15], formerly the International Obesity Task Force (IOTF), recommend age- and sex-specific BMIs with which to assess weight status at ages 2 to 19 years. The most common forms of its use are the Z-score and percentiles. However, different studies have shown that these international references provide different results in the prevalence of overweight and obesity in children and adolescents [16,17,18,19,20].

Therefore, considering the different results between the various international references, the objective of this research was to determine the level of agreement of the prevalence of underweight, normal weight, overweight, and obesity estimated with three international references in a population of Mexican children and adolescents between 2 and 18 years of age using specific BMI sections by sex and age.

## 2. Materials and Methods

### 2.1. Sample and Study Design

The sample consisted of children and adolescents between the ages of 2 and 18 (24–216 months). The information from the database of the National Health and Nutrition Survey (ENSANUT, Mexico City, Mexico) Continua 2022 [21] was analyzed. This survey has national, probabilistic, multistage, and stratified data, which allows for national representativeness. The anthropometry questionnaire was used. The design of this study was cross-sectional.

### 2.2. Body Mass Index (BMI)

The anthropometric measurement questionnaire was used to obtain weight and height measurements of children and adolescents. From these measurements, the BMI was calculated using BMI = kg/m^2^, where kg is a person’s weight in kilograms, and m^2^ is height in meters squared. The goal was to establish the cut-off points of the different measurements in children and adolescents recommended by the WHO [11,12], CDC [14], and IOTF [15].

#### 2.2.1. Exclusion and Inclusion Criteria

All children and adolescents from the ENSANUT Continua 2022 database were included, with weight, height, and age records.

#### 2.2.2. BMI Z-Score—WHO

The WHO recommends measuring nutritional status in those over 2 years of age and under 19 years of age through the Z-score (standard deviation) of the BMI for age [12,13]. For children under 19 years of age, the following criteria were established: less than −2.00 standard deviations is considered underweight; between −2.00 and +0.99 standard deviations is considered normal weight; between +1.00 and +1.99 standard deviations is considered overweight; and greater than or equal to +2.00 deviations is considered obese.

#### 2.2.3. BMI Percentile—CDC

The CDC established BMI percentile charts by sex and age that indicate the growth pattern of children and adolescents [14]. The BMI categories by age and their percentiles were established as follows: below the 5th percentile (P5) indicates underweight; between the 5th percentile and below the 85th percentile (P85) indicates normal weight; between P85 and below the 95th percentile (95P) indicates overweight; and a BMI above or equal to the 95th percentile indicates obesity.

#### 2.2.4. IOTF Cut-Off Points

For sex- and age-specific cut-off points defined by the International Obesity Task Force (IOTF) [15], the points pertain to ages between 2 and 18 years, based on a BMI at 18 years corresponding to an adult.

### 2.3. Statistical Analysis

The results obtained from the prevalence of the three references were analyzed with the χ^2^ (chi-square) test, with a significance level of *p* < 0.05. This analysis was performed using IBM SPSS Statistics (Version 30, Armonk, NY, USA).

### 2.4. Cohen’s Kappa Statistics

For the analysis of the data, the level of agreement was established by applying Cohen’s Kappa statistics, considering the following cut-off points [22]: slight (0.0–0.20); fair (0.21–0.40); moderate (0.41–0.60); substantial (0.61–0.80); and almost perfect (0.81–1.00) [23].

## 3. Results

### 3.1. Total Children and Adolescents

The information of 5108 children and adolescents was analyzed, 2492 from the female group (48.8%) and 2616 from the male group (51.2%), with a greater number of children under 10 years of age (2870 children and adolescents). The distributions are shown in Table 1.

### 3.2. BMI Classification According to International Reference

Generally, the BMI categorizes individuals into four classifications: underweight, normal weight, overweight, and obese. However, it is important to note that individual variations can occur, and the BMI alone is not sufficient for accurately determining if someone is obese or malnourished. In children, the BMI provides a useful comparison among peers of the same sex and age. For this population, a BMI below the 5th percentile is classified as underweight, while a BMI above the 95th percentile is considered obese. It is very important to consider that elevated muscle mass and weight falsely increase the BMI in certain individuals. The BMI number and classifications by sex and age are described below [24].

#### 3.2.1. BMI Classification by Sex

According to the weight classification, the prevalence of low weight in children and adolescents was higher when applying the IOTF reference, based on the CDC reference. In the cases of normal weight, overweight, and obesity, prevalence was higher when the WHO classification was used, and the prevalence of the normal weight and overweight categories was lower if the CDC reference was considered (Table 2).

When the results are analyzed by sex, the prevalence of underweight, in both sexes, is higher when the IOTF reference is applied. For the normal weight category, the prevalence is higher in both cases when the CDC reference is used (Table 2). In the cases of overweight and obesity, in both, the prevalence is higher when the WHO reference was used, the lowest prevalence of overweight was obtained when the CDC reference was considered, and a lower prevalence of obesity was obtained when the IOTF reference was considered, as can be seen in Figure 1. In general terms, the female group showed a higher prevalence of overweight than the male group; opposite results were obtained in the prevalence of obesity.

#### 3.2.2. BMI Classification by Age

Regarding age, the prevalence of overweight in the female group was higher between 2 and 8 years when the WHO reference was used, and lower for all ages when the CDC reference was applied. The male group showed similar overweight trends at all ages when the WHO and CDC references were used; that is, the prevalence of overweight was lower when the IOTF reference was used (Figure 2). In the case of obesity prevalence in the female group between 2 and 9 years old, it was higher when the WHO reference was used, and a lower prevalence was observed in the group between 5 and 18 years of age when the CDC reference was applied. For the male group, the prevalence of obesity was lower when the IOTF reference was used; similar trends in the prevalence of obesity were obtained with the WHO and CDC references. The results are shown in Figure 2.

### 3.3. Level of Agreement

Table 3 shows the level of agreement according to sex and age. When the analysis was carried out by sex, the result was substantial and almost perfect agreement. In the case of the female group, the level of agreement was considerable when the WHO-IOTF reference was compared and almost perfect when the WHO-CDC and CDC-IOTF references were compared. For the male group, moderate agreement was obtained when the WHO-IOTF reference was compared, considerable when the CDC-IOTF reference was compared, and almost perfect agreement when the WHO-CDC reference was compared.

When the analysis was carried out by age, the level of agreement was lower in both sexes. The same results were found in the age group of 2 to 4 years of age; however, the lowest level of agreement was found when comparing the WHO-IOTF reference. The highest level of agreement was obtained in the age group of 12 to 18 years of age, with an almost perfect level. The results are shown in Table 3.

## 4. Discussion

The prevalence of childhood and adolescent obesity has increased in most countries, so joint policy initiatives across government departments are needed to develop and implement interventions focused on obesity prevention [25].

Mexico ranks first in childhood obesity worldwide. This represents a public health problem because overweight/obesity in this age group will probably contribute to obesity for the rest of their lives and increase morbidities related to this disease [26]. One of the factors that explains this disease is the place of birth or residence. The prevalence of obesity is higher in Western countries, and the place of birth or residence has been linked with eating habits [27]. Other factors related to overweight and obesity in schoolchildren and adolescents are the presence of excess weight in the mother, increased time in front of a screen, and reduced time for physical activity [28].

Therefore, it is important to highlight that when Mexican children and adolescents are evaluated, the prevalence of weight categories is differentiated according to the international reference used, showing a greater difference between the WHO and the IOTF, mainly at low weights. In this category, there was a total difference of 7.48% (*n* = 382) between the references, with a higher percentage of low weight in the IOTF. The same trend occurred between the female (7.33%; *n* = 197) and male (7.25%; *n* = 185) groups. In the case of normal weight, the three classifications showed similar results, with a total difference of 2.23% (*n* = 114) between the CDC classification (lower) and the WHO classification (higher). The difference in the female group was 4.01% (*n* = 100) between the CDC reference and the IOTF, and in the male group, a difference of 2.33% (*n* = 59) between the CDC and WHO references was observed. For overweight, the WHO reference indicates a higher percentage in this weight category compared to the CDC, with a difference of 2.19% (*n* = 246). If analyzed by sex for these same references, it was found that the greatest difference occurred in the male group with a difference of 4.97% (*n* = 130) compared to the female group. For the obesity category, the WHO classification obtained a greater difference of 5.2% (*n* = 266) compared to the IOTF classification, and by sex, the percentage difference showed the same trend between the WHO and the IOTF references. However, the percentage of obesity was higher in the male group regardless of the international reference used.

Similar results were found in the work of Padula and Selceda [19], with a study population of 737 healthy Argentine children aged 2 to 5 years, which found that the prevalence of overweight and obesity differed significantly between using the IOTF and CDC references. They estimated that the prevalence of obesity is lower if the IOTF tables are applied, with a difference of 11.1% in the female group and 14.5% in the male group, mainly at the age of 4 years. Similar results were found in the study by Carrillo et al. [29], which analyzed 634 children and adolescents between 6 and 17 years of age, in which the prevalence of overweight and obesity was lower when the IOTF reference was used (7.9% in males and 7.1% in females). They found that the prevalence of overweight is higher if the OITF reference is used (31.1% men; 28.8% women), and in the case of obesity, the prevalence is higher if the OITF reference is used (31.1% men; 28.8% women). Wang and Wang [30] obtained similar results in populations of children and adolescents aged 6 to 18 years in the United States, Russia, and China, with a lower prevalence of obesity when using the IOTF benchmark compared to the WHO benchmark. Similar results were obtained in the research by Bergel et al. [20], carried out in Venezuela and Spain, where the prevalences of overweight (Venezuela = IOTF: 11.7%; WHO: 16.0%; Spain = IOTF: 24.4%; WHO: 27.5%) and obesity (Venezuela = IOTF: 3.9%; WHO: 6.3%; Spain = IOTF: 3.8%; WHO: 10.5%) were lower in adolescents between 10 and 13 years old when using the IOTF reference compared to the WHO reference.

On the other hand, when analyzing the level of concordance using the Kappa statistic, the study by Granado et al. [31] found that the level of concordance of the BMI for children and adolescents in Paraguay between 5 and 16 years was 0.733 for the male group and 0.452 for the female group, showing the lowest level of agreement in this group. In the research of Wang and Wang [30], the level of agreement for the prevalence of overweight and obesity among the WHO-IOTF references for children and adolescents between 6 and 18 years of age in China, Russia, and the USA was calculated. They found that in China, the level of agreement of overweight in children from 6 to 9 years of age was 0.92 (men = 0.92; women = 0.92); in Russia, it was 0.90 (men = 0.86; women = 0.95); and in the USA, it was 0.90 (men = 0.80; women = 0.90). In the case of adolescents aged 10 to 18 years, the level of agreement in China was 0.94 (men = 0.98; women = 0.90); in Russia, it was 0.88 (men = 0.92; women = 0.84), and in the USA, it was 0.92 (men = 0.93; women = 0.92). Therefore, the best level of agreement was obtained in male adolescents in China, and the worst, in male and female adolescents in Russia. In analyzing the level of agreement between the WHO-CDC references for height-for-age measurements, Rincón et al. [32] found that in children under 17 years of age (*n* = 31,961) in Caldas, Colombia, the level of agreement was lower in the female group (0.754) compared to the male group (0.829). The results indicated that there is a lower level of agreement between the WHO-IOTF references, which is in agreement with our results.

This study’s limitations include potential inconsistencies in the measurement of height and weight in the database. Some populations in the country could have a higher proportion of muscle mass or excessive fat distribution around the abdominal area that does not align with the classifications assessed, which may result in the misclassification of individuals’ weight status. Therefore, the presence of abdominal obesity, also known as central or visceral obesity, could affect the overall prevalence estimates. The rise of 99% in the likelihood of increased cardiometabolic risk in children and adolescents with overweight, independent of age and sex, at the nationwide scale is alarming. Abdominal obesity has been consistently linked to cardiometabolic risk factors including insulin resistance, type 2 diabetes, hypertension, and dyslipidemia, among other early markers of cardiovascular disease. Thus, it is necessary to include the measurement of abdominal obesity in the determination of nutritional status and to identify children and adolescents with cardiometabolic risk [33]. Other biases arise from the fact that the information collected in the ENSANUT is based on self-reporting by participants or their parents/guardians and may exclude certain groups of children and adolescents from rural areas that are very difficult to access or that are in situations of extreme vulnerability.

One of the strengths of this study is that, so far, there has been no comparison between the prevalence of underweight, normal weight, overweight, and obesity using the WHO, CDC, and IOTF reference cut-offs of Mexican children and adolescents. The sample size was also important. The information on these age groups comes from analysis units (households, health service users, and age groups) that are representative not only at a national level but also at a regional level. ENSANUT Continua 2022 collected information from 13,879 households, with 10,160 completed household interviews [34]. In addition to the importance of calculating the level of concordance between the different international references that determine the nutritional status of children and adolescents, it is relevant to carry out future analyses of the relationship between the BMI and body fat and between the BMI and the risk of developing chronic degenerative diseases at earlier ages due to the increase in the prevalence of non-communicable chronic diseases in the Mexican population [35].

Mexico is within the group of countries with high obesity rates, such as Chile and Argentina (https://data.worldobesity.org/#MX|36|C|F, accessed on 24 January 2025). The childhood obesity rate in Mexico, according to ENSANUT Continua 2022, shows a slight decrease compared to previous years; however, it is still very high. These countries must implement comprehensive strategies to address this problem and promote healthy lifestyles in children and adolescents.

## 5. Conclusions

Based on the data analyzed, it can be concluded that the prevalence of weight categories in Mexican children and adolescents between 2 and 18 years of age differs between the international references used, showing the lowest concordance between the WHO reference and the IOTF reference. Therefore, the weight category depends on the reference used, so each one should be used with caution since the results are used to inform actions of prevention, surveillance, and the control of nutrition in childhood and adolescence for the timely detection of chronic health problems and effects of social deficiencies.

In addition, future studies should be considered such as for obtaining a criterion for Mexico. However, this has advantages and disadvantages. Advantages include greater precision and relevance to the Mexican context, for example, for indigenous populations [36], including body composition and genetic factors for more specific interventions. Difficulties include having the resources to develop and validate the criteria, and implementing and comparing the new criterion with international standards would be more complex.

## Figures and Tables

**Figure 1 nutrients-17-00587-f001:**
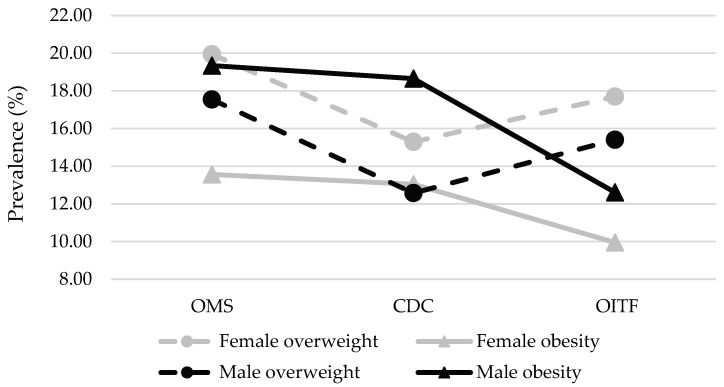
Comparison of the prevalence (%) of overweight and obesity in the population aged 2 to 18 years, Mexico 2022.

**Figure 2 nutrients-17-00587-f002:**
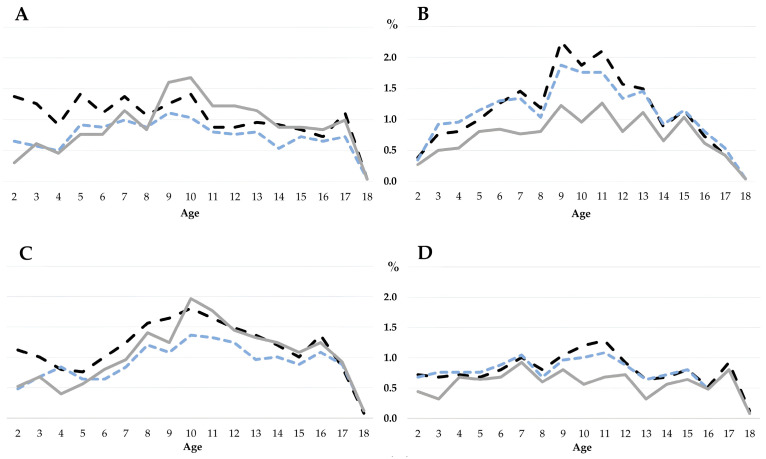
Comparison of the prevalence (%) of overweight and obesity by sex and age in Mexican children and adolescents. (**A**) Female overweight, (**B**) male overweight, (**C**) female obesity, (**D**) male obesity. The black line is the WHO reference, the blue line is the CDC reference, and the gray line is the IOTF reference.

**Table 1 nutrients-17-00587-t001:** Distributions by sex and age of children and adolescents, Mexico 2022.

Age	Female	Male	Total	%
2	157	167	324	6.34
3	199	168	367	7.18
4	190	203	393	7.69
5	160	165	325	6.36
6	157	183	340	6.66
7	180	217	397	7.77
8	157	177	334	6.54
9	191	199	390	7.64
10	165	179	344	6.73
11	155	163	318	6.23
12	148	166	314	6.15
13	131	149	280	5.48
14	122	139	261	5.11
15	127	126	253	4.95
16	118	109	227	4.44
17	119	101	220	4.31
18	16	5	21	0.41
Total	2492	2616	5108	100.00

**Table 2 nutrients-17-00587-t002:** Weight status in the population aged 2 to 18 years using three international references, Mexico 2022.

		Weight Status of Study Participants *n* (%frequencies)			
Sex	International Reference	Underweight	Normal	Overweight	Obesity
Female $(\chi^{2}=$ 172.915; *p* value < 0.05)	WHO	49 (1.97)	1608 (64.50)	497 (19.94)	338 (13.56)
	CDC	129 (5.18)	1657 (66.49)	381 (15.29)	325 (13.04)
	IOTF	246 (9.87)	1557 (62.45)	441 (17.69)	248 (9.95)
Male $(\chi^{2}=$ 181.191; *p* value < 0.05)	WHO	57 (2.18)	1594 (60.93)	459 (17.55)	506 (19.34)
	CDC	144 (5.50)	1655 (63.26)	329 (12.58)	488 (18.65)
	IOTF	242 (9.25)	1641 (62.73)	403 (15.41)	330 (12.61)
Total $(\chi^{2}=$ 349.437; *p* value < 0.05)	WHO	106 (2.08)	3202(62.69)	956 (18.72)	844 (16.52)
	CDC	273 (5.30)	3312 (64.80)	710 (13.90)	813 (15.92)
	IOTF	488 (9.55)	3198 (62.61)	844 (16.52)	578 (11.32)

**Table 3 nutrients-17-00587-t003:** Level of agreement by Kappa coefficient between the different international references.

Sex/Age Group	WHO-CDC	WHO-IOTF	CDC-IOTF
Female	0.812 *	0.682 *	0.816 *
2–4 years	0.740 *	0.431 *	0.645 *
5–11 years	0.796 *	0.693 *	0.845 *
12–18 years	0.869 *	0.785 *	0.861 *
Male	0.800 *	0.598 *	0.756 *
2–4 years	0.659 *	0.346 *	0.601 *
5–11 years	0.803 *	0.571 *	0.750 *
12–18 years	0.864 *	0.768 *	0.842 *
Total	0.806 *	0.639 *	0.785 *

* *p* value ≤ 0.05.

## Data Availability

The original contributions presented in this study are included in the article. Further inquiries can be directed to the corresponding authors.

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
