# Peer review of "Agreement on the Prevalence of Body Mass Index (BMI) in Mexican Children and Adolescents Using Different International References"

_nutrients, 2025, doi:10.3390/nu17030587_

Round 1
Reviewer 1 Report
Comments and Suggestions for Authors
This research article deals with an interesting topic. However, some points should be addressed.
- Subheadings (Background, Methods, Results, Conclusions) should be added in the Abstract.
- The sentence in lines 31-36 is very complex and it should be rephrased to be more easily readable from the readers.
- Epidemiological data concerning overweight and obesity in mexican children and adolescents should be added in the 1st paragraph of the Introduction section.
- In section, the exclusion and inclusion criteria for children and adolescents enrollment should be reported.
- In section 2.2, Please, be more specific if the weight anf height of the study population are self-reported by questionnaires or are measured data.
- The resolution of Figure 2 should be improved.
- The sentence in line 191-194 should be split into two smaller sentences.
- In the 1st paragraph of the Discussion section, the authors should add some statements conserning the reasons for the high prevalence of overweight and obesity in Mexican children and adolescents.
- The paragraphs in lines 195-213 should be merged into one paragraph.
- The sentence in lines 227-228 "The same results in the prevalence of overweight 227 and obesity were found in other studies that applied the study references." needs more analysis by reporting additional studies.
- The paragraph in lines 229-235 needs more analysis by reporting additional studies.
- In the paragraph with the limitations of the study the authors report the fat distribution. In this aspect, the authors should also discussed the phenomenon of the increased prevalence of abdominal obesity in children and adolescents wordwide.
- Another strength of the study that could be reported is the relative high numebr of the enrolled children and adolescences.
- The authors should report their critical opinion if the analysis of their studies should be applied separately in children from adolescents.
- In the Conclusion section, the authors should proposed what future studies could be done based on their results.
Reviewer 2 Report
Comments and Suggestions for Authors
1. In 2.3 section, statistical analysis, the χ2 (chi-square) tests were not found using in all table or figure, even in the content. Were the authors sure to use the χ2 (chi-square) tests in the manuscript? If they were surely used, please also report the χ2 values and their p-values.
2. The study had a finding of disagreement of the three references and suggested the three references should be used cautiously. In my opinion, the disagreement among the three references seems predictable without doing this study. The authors should offer some standards to judge how large in percentage of disagreement should be considered severe and then to do something, not just suggest that they be used cautiously.
3. Lines 99-100, “the BMI (weight/height2) was calculated.” It might be misunderstood. It was better also to express with wordings, such as “weight is divided by squared height.”
4. With ”Cohen's Kappa index was used.“ for agreement, it had better to present the significance test results and p-values in Table 3.
5. The manuscript needed to offer more important contribution, not just “comparison between the prevalence of underweight, normal weight, overweight, and obesity using the WHO, CDC, and IOTF reference cut-offs of Mexican children and adolescents”
Round 2
Reviewer 1 Report
Comments and Suggestions for Authors
The authors have significantly improved their manuscript.
Reviewer 2 Report
Comments and Suggestions for Authors
OK